# ADGR: Admixture-Informed Differential Gene Regulation

**DOI:** 10.3390/genes14010147

**Published:** 2023-01-05

**Authors:** In-Hee Lee, Sek Won Kong

**Affiliations:** 1Computational Health Informatics Program, Boston Children’s Hospital, Boston, MA 02215, USA; 2Department of Pediatrics, Harvard Medical School, Boston, MA 02115, USA

**Keywords:** regulatory elements, local ancestry, gene expression, genotype-tissue expression

## Abstract

The regulatory elements in proximal and distal regions of genes are involved in the regulation of gene expression. Risk alleles in intronic and intergenic regions may alter gene expression by modifying the binding affinity and stability of diverse DNA-binding proteins implicated in gene expression regulation. By focusing on the local ancestral structure of coding and regulatory regions using the paired whole-genome sequence and tissue-wide transcriptome datasets from the Genotype-Tissue Expression project, we investigated the impact of genetic variants, in aggregate, on tissue-specific gene expression regulation. Local ancestral origins of the coding region, immediate and distant upstream regions, and distal regulatory region were determined using RFMix with the reference panel from the 1000 Genomes Project. For each tissue, inter-individual variation of gene expression levels explained by concordant or discordant local ancestry between coding and regulatory regions was estimated. Compared to European, African descent showed more frequent change in local ancestral structure, with shorter haplotype blocks. The expression level of the Adenosine Deaminase Like (*ADAL)* gene was significantly associated with admixed ancestral structure in the regulatory region across multiple tissue types. Further validations are required to understand the impact of the local ancestral structure of regulatory regions on gene expression regulation in humans and other species.

## 1. Introduction

The known history of human evolution and migration out of Africa, and the recent migration of people across the continents, suggest that the genomes of modern people are a composite admixture of haplotypes from multiple ancestral populations [1]. The admixture of haplotype blocks could locally introduce novel combination of alleles not observed in ancestral populations [2]. Admixture can be thought of as a series of meiotic recombination over multiple generations that contribute to genetic diversity, as successive offspring exhibit admixed genomes with new combinations of alleles [3]. As such, recombination and mutation are the main sources of genetic variation in populations. Studying the genomic composition of admixed individuals across diverse populations provides a lens to infer the recombination rate of recent admixture events estimated by constructing genetic maps using pedigree or linkage disequilibrium (LD) based approaches [4].

Genome-wide association studies (GWASs) have revealed that most disease-associated risk loci lie outside of protein coding genes [5]. Fine-mapping and expression quantitative trait loci (eQTL) analysis demonstrated that risk alleles in the non-protein coding region may alter the regulation of gene expression by modifying the binding affinity of diverse DNA binding and interacting proteins implicated in gene expression regulation [6]. For instance, a single nucleotide polymorphism (SNP) changes the binding affinity of transcription factors and epigenetic regulators, which results in differential efficiency in transcription [7] and mRNA processing [8]. Moreover, the transferability of GWAS findings to other populations is challenging since variant allele frequency of risk alleles varies between populations. To this end, estimating the rate of recent recombination events allows for the identification of genomic loci that may be associated with disease across populations [1].

Almost 80% of currently available GWASs were performed using DNA samples from European descent, especially from the people of the United States, the United Kingdom and Iceland [9,10,11]. Therefore, previous eQTL studies used genotype-derived global ancestry and/or top-most principal components of genotypes as covariates, or just focused on minimally admixed European populations [12,13]. Generalizing the findings from these studies to non-European populations is challenging, especially for admixed populations. To this end, Zhong and colleagues incorporated local ancestry information to explain a proportion of variance in gene expression levels between individuals and found polygenic contributions to gene expression variations in admixed individuals [14]. Thus, a new method must be sought out that not only describes the effect of local ancestry on gene expression regulation, independent of population, but also adjusts the model to reduce false positive association between genotype and expression phenotype.

Our approach to describing admixed ancestral structure is focused on transitions—i.e., genomic loci delineating potential recombination events between continent-level populations—that indicate changes in local ancestry from one ancestral population to another. We inferred local ancestry with whole genome sequencing (WGS) data using RFMix, an algorithm that learns from a reference panel of haplotypes and genetic recombination maps to infer the most likely local ancestral structure of a query genome. RFMix uses a genetic map-based approach for estimating local ancestry, which differs from the other algorithms based on linkage disequilibrium (LD) and, therefore, is not bound to the limitation of classifying local ancestry of up to two populations in LD-based approaches [15]. Thus, we could use larger reference panels to predict among several ancestral populations at a time. We propose a method, Admixture-informed Differential Gene Regulation (ADGR), for modeling differences in gene expression, which may be used as a proxy for phenotypic changes associated with disease, due to changes in local ancestry between protein coding and upstream regulatory regions as a result of admixture.

## 2. Materials and Methods

We collected paired genome-wide variant calls from phased WGS and tissue-wide RNA-seq datasets from the Genotype-Tissue Expression (GTEx) project (release V8) [16,17]. For 838 phased WGS variant call files (VCFs), we used RFMix (version 2) to infer genome-wide local ancestral structure [2]. Reference panels were constructed from combinations of continent-level populations of the 1000 Genomes Project: African (AFR), Admixed American (AMR), East Asian (EAS), European (EUR), and Southeast Asian (SAS). Samples that represented each continent-level population were selected from populations that showed the least degree of admixture: Yoruba (YRI) for AFR, Peru (PEL) for AMR, Han Chinese (CHB) for EAS, Utah Residents with Northern and Western European ancestry (CEU) for EUR, and Italian Telugu (ITU) for SAS. From each of the five populations, we chose 85 individuals with lesser degrees of admixture according to ADMIXTURE results [18]. Three reference panels that consisted of two-, three-, and five-populations were used to assign local ancestry with RFMix. A two-population panel consisted of the 85 individuals from each of YRI and CEU, and a three-population reference panel consisted of individuals from YRI, PEL, and CEU.

For each individual WGS from GTEx, RFMix assigned one of the ancestral populations in the reference panel to each of two alleles along the chromosome. Then, consecutive regions with the same local ancestry formed a haplotype block. A transition point was defined as the genomic locus between two adjacent haplotype blocks with different local ancestries. For further analysis, we focused on two populations—i.e., EUR (N = 715) and AFR (N = 103)—since there were only a small number of individuals from the other populations (AMR N = 2, ASN N = 12 and unknown N = 6) in the GTEx project.

We defined protein coding and upstream regulatory regions according to the GENCODE annotation (version 26) [19,20]. For each protein coding gene, the upstream genomic region from the transcription start site (TSS) was further partitioned to three regions by distance from TSS: immediate upstream (up to 5 kilobase pairs, kbps), distant upstream (5–50 kbps) and distal (50–500 kbps) regions (Figure 1).

For each gene *i*, we compared two linear regression models with and without the presence of transition points in upstream regions. The baseline model (M_0_) was *y_i_~age + sex + global ancestry* and the alternative model (M_1_) was *y_i_~trans + age + sex + global ancestry*, where *y_i_* denoted the standardized expression level of gene *i*. For global ancestry, we cross-checked reported information in the GTEx phenotype table and predicted global ancestry derived from WGS—the largest proportion of local ancestries for an individual by RFMix. For prostate, uterus, and ovary, we excluded the variable *sex* from both M_0_ and M_1_. The independent variable *trans* represents the transition point status upstream of the gene *i*. We used three different approaches to model the local ancestry transition point: (1) dominant model: *trans* = 1 if any of the two alleles contained transition points in the upstream of the gene (otherwise *trans* = 0); (2) additive model: the variable *trans* equals the number of alleles that have transition points; and (3) recessive model: *trans* = 1 only if both alleles have transition points (otherwise *trans* = 0). For each model and three upstream regions, we compared the two models—i.e., M_0_ and M_1_ using a two-sided *χ*^2^ test—to find the genes for which expression levels were significantly better explained by the presence of transition events in upstream regions.

## 3. Results

### 3.1. Local Ancestral Structure of 838 Individuals

The reported global ancestry of each individual from GTEx phenotype data matched with the ancestral population predicted for the largest portion of its genome for all individuals in the current study. Global and local ancestral structure were summarized in three ways. Firstly, we created an ADMIXTURE-style graph to visualize the overall proportion of continent-level populations for all. Secondly, we counted the total number of transition points in each individual. Thirdly, we checked the distributions of putative haplotype block sizes between two transition points. In Figure 2A–C, the two largest populations in the GTEx project—i.e., African and European—are shown, and the largest proportion of predicted local ancestry from RFMix was concordant with the reported global ancestry from the GTEx phenotype metadata. This observation was consistent regardless of the number of ancestral populations in the reference panels for RFMix: two (AFR and EUR, Figure 2A), three (AFR, AMR and EUR, Figure 2B) or five populations (AFR, AMR, ASN, EUR, and SAS, Figure 2C).

Using the five-population reference panel, African individuals had 2592 transition points on average in their genome (standard deviation (SD) 846, range from 530 to 4914) compared to the average of 4758 transition points found in Europeans (SD 213, range from 4404 to 6120). However, with the two-population reference panel including only AFR and EUR, the average number of transition points in Europeans was significantly reduced compared to reference panels with three or five populations (Figure 2D–F). European individuals showed a higher increase in the number of transition points because about 20% of genomes, which were mapped to EUR with the two-population reference panel (AFR and EUR), were mapped to non-EUR populations as the reference panel changed (right panels in Figure 2A–C). However, for African individuals, the majority of the genome was consistently mapped to AFR and only small portion (~5%) of the genome was mapped differently as the reference panel changed (left panels in Figure 2A–C).

The distribution of block lengths is shown in Figure 2G–I. Specifically, the blocks shown here are for the stretches of putative local ancestry that were concordant with the reported global ancestry in each individual. In both population groups, the average block length predicted by RFMix was approximately 350 kbps. In the five-population panel, African subjects on average have longer stretches of concordant ancestry, as shown by the heavier tail on the right side of the distribution. In the two-population, on the other hand, Europeans showed a similar trend, with a right-side heavy tail in the distribution.

### 3.2. Transition of Local Ancestral Structure and Gene Model

We checked the location of potential transitions between ancestral blocks in relation to the gene definitions according to the GENCODE annotation. Here, we focused on the results generated using the two-population reference panel. Out of 1413.5 (SD 423.92) total transitions per individual, African individuals had 830.3 (SD 246.65) transitions in the intergenic region. European individuals had 82.8 (SD 103.83) transitions in the intergenic region out of 137.7 (SD 177.87) total transitions on average (Figure 3).

In both groups, ~60% of total transitions were observed in the intergenic region and 40% in the genic region. Most transitions in the intergenic region were within 500 kbps from TSS, with the largest number between 50 kbps and 500 kbps (center box in Figure 3). The rightmost box in Figure 3 shows that most transitions in the genic region were within introns or the untranslated region (UTR), leaving only a few in exons: 45.6 (SD 14.23) transitions in exons out of 583.2 (SD 179.51) in the genic region among African individuals, and 4.5 (SD 5.96) transitions in exons out of 54.9 (SD 74.43) in the genic region among European individuals.

### 3.3. Gene Expression Levels Associated with Admixed Ancestral Structure in the Regulatory Region

Next, we used RFMix predictions using the two-population reference panel and standardized gene expression levels to rank the genes whose expression levels could be explained by the presence of transition events in the upstream regions. The gene expression matrices for available tissue types were downloaded from GTEx single-tissue cis-eQTL data in the GTEx portal (https://gtexportal.org, accessed on 26 August 2019). As shown in Table 1, most of the candidate associations were found for genes with transitions in 50~500 kbps upstream. The list of candidate genes varied by tissue types; however, the *ADAL* gene encoding the adenosine aminase like protein (ADAL) was significantly associated with admixed ancestral structure in the regulatory region, especially in 50~500 kbps upstream, across multiple tissue types for both dominant and additive models (Table 1).

For the individuals with transitions in 50~500 kbps upstream of *ADAL*, expression levels of this gene were significantly lower compared to the individuals without transition in the upstream (Figure 4). Interestingly, *ADAL* was differentially expressed between African Americans and European Americans with colorectal cancer [21].

Compared to dominant or additive models, we found significantly smaller numbers of candidate genes with recessive models, which suggested that most of the transitions in the upstream were heterozygous. Interestingly, *ADAL* was consistently found significant in multiple tissue types for both dominant and additive models. We also observed that all the candidate genes from the additive model had transitions in only one of alleles (heterozygous). Since transition was, after all, infrequent, we expected that homozygous transition would be very rare, which made it difficult to find candidate genes using a recessive model. We found only one candidate gene with the recessive model (*SH3GLB1* in uterus).

### 3.4. Gene Expression Levels in Chromosome 8q24 Associated with Local Ancestral Transition between Africans and Europeans

Prostate cancer is one of the most common malignancies among men in the U.S., and the incidence among African Americans is ~1.6-fold higher compared to European Americans. Freedman and colleagues used a whole-genome admixture scan to discover susceptibility loci for prostate cancer in African Americans and found chromosome 8q24 as a significant risk locus for prostate cancer, especially for African descent. However, candidate genes in 8q24 were not identified [22]. We focused on the transition events and genes in chromosome 8q24 to find the candidate genes that were differentially regulated by admixed ancestral structure in regulatory regions. We found that more significant associations were from recessive models, in contrast to the whole genome analysis in the previous section (Table 1). Trafficking protein particle complex 9 (*TRAPPC9*) and pyrroline-5-carboxylate reductase (*PYCRL*) were significantly associated with ancestral admixture in the regulatory region across multiple tissue types (Table 2). *TRAPPC9* is implicated in tumorigenesis through the NF-kB signaling pathway [23]. *PYCRL* plays a role in proline biosynthesis [24] and was significantly associated with prostate proliferation in a murine model of prostate cancer [25]. We checked the locations of the genes and their regulatory regions in Table 2 and did not find any overlap of distal regulatory regions between the genes that were significant in a tissue type (Figure 5).

## 4. Discussion

The proportion of phenotype variance explained by genotype is relatively small for many human traits, including diseases [26]. Gene expression could be used as an endophenotype that is a mediator between genotype and phenotype. Indeed, genetic variants have larger effects on the variance in CpG DNA methylation and gene expression levels compared to effect sizes on phenotypic variance [27]. Positive findings from eQTL analysis and fine-mapping of GWAS results, as well as statistical methods such as PrediXcan [28] and fusion [29], suggest that inter-individual variation of tissue-specific gene expression could be explained by from a single SNP to genome-wide genotype of an individual [30]; however, it is likely that multiple genetic variants in the regulatory region contribute to differential gene expression regulation across individuals [31]. 

On average, the number of putative transition points was smaller in European descent compared to African descent when a two-population reference panel was used. This observation is consistent with previous reports regarding greater genetic diversity in Africans with shorter LD-block sizes [32,33,34]. However, we found more frequent transition points in Europeans with three- or five-population reference panels, which was likely due to the limitation of the algorithm in assigning local ancestry to one of the populations in a population reference panel. RFMix performs better with individuals from complex admixed populations compared to the other methods [35]; however, the subjects enrolled in the GTEx project were not necessarily from complex admixed populations. Therefore, the results with two-population reference panel were consistent with previous reports as to the number of transition points and block sizes in Europeans and Africans.

The association between the local ancestry transition In the distal upstream (50~500 kbps from TSS) of *ADAL* and its expression level was recurrently observed across 13 tissue types. ADAL has an important role in the metabolism of mRNA across cell types in multiple species. N^6^-methyl adenine (m^6^A) is the most abundant posttranscription modification of mRNA, and m^6^A is turned over to generate N^6^-mAMP. ADAL is an evolutionary conserved catalytic enzyme that hydrolyzes N^6^-methyl-AMP (N^6^-mAMP) to produce inositol monophosphate (IMP) and methylamine [36]. Therefore, differential regulation of *ADAL* could have an impact on mRNA stability and metabolism [37]. In our analysis, *ADAL* was significant in multiple tissue types, which was not solely due to the ubiquitous expression of *ADAL*. Indeed, 77% of all tested genes (N = 13,556) were quantitatively measured in 40 or more tissue types in GTEx. Nonetheless, *ADAL* and *HLA-DQB2* were two genes that were significantly associated with the local ancestry of regulatory regions in diverse tissue types at a study-wide statistical threshold of 1.87 × 10^−8^ = 0.05/2,670,793 (the number of all tests for all genes across available tissues, models of transition loci (additive/dominant/recessive), and distance between transition loci and TSS (immediate/distant/distal)). Given the sample size in GTEx data (N = 838), however, it requires replication study with a larger sample size for validation.

Gene expression levels are influenced by both genetic and environmental factors. Moreover, environmental factors such as lifestyle and diet are often linked with an individual’s global ancestry. In the current study, we aimed to delineate the effect of genetic variants in regulatory regions, in aggregate, on the inter-individual variation of gene expression levels. For some genes, mean expression levels could be different between populations due to environmental factors and gene-environment interactions. In the current study, gene expression levels were residualized for global ancestry (along with sex and age) to estimate the variance explained by the change in local ancestry between regulatory and coding regions. Therefore, the genes that were significantly differentially expressed between populations might have been missed in our analysis.

Although our approach identified candidate genes that may be differentially expressed due to the discordant local ancestry of the regulatory region compared to coding regional structure, there are several technical challenges that make interpretation difficult. Firstly, the potential impact of allele-specific expression was not explored [38]. Most transition events were heterozygous in our dataset. Thus, one of two alleles with discordant local ancestry in regulatory regions could have a differential effect on gene expression. The next generation sequencing technique used to generate WGS and RNA-seq data for the GTEx project has limitations in resolving the haplotype of regulatory regions relative to coding regions. Third generation long-read sequencing techniques, such as 10× linked-reads sequencing and Oxford Nanopore, would enable the generation of an accurate allele specific map of regulatory and coding regions [39]. Secondly, transition events in upstream might have different impacts across cell types, which it was not possible to analyze using bulk RNA-seq data from the GTEx project. Thirdly, there was a lack of reliable reference haplotype data in the latest human genome build GRCh38. RFMix requires prior information from a human genome-wide recombination map and a reference panel of different ancestral populations matching the target population. Therefore, local ancestry prediction with RFMix is dependent on the quality and size of the required materials. However, publicly available genomic data are biased with European populations [9,40], limiting our ability to investigate individuals of non-European ancestry.

DNA double strand break sites—i.e., sites of meiotic recombination—are often determined by PR domain-containing protein 9 (PRDM9) in the human [41]. Interestingly, different ancestral populations have distinct recombination hotspots. Moreover, *PRDM9* alleles and DNA sequence motif binding PRDM9 show difference between Europeans and African Americans [34]. High resolution genetic maps for diverse ancestral populations are not readily available yet. As such, we found significant differences in the number of transitions and the distribution of size of haplotype blocks between the results using three different reference panels. Further refinement of local ancestry prediction methods would improve the statistical power to detect gene expression variation explained by admixed ancestral structure in the regulatory region. 

## 5. Conclusions

In the current study, we illustrated an intuitive way to estimate the impact of local ancestry on gene expression levels in the two populations (i.e., AFR and EUR) using GTEx WGS data. A total of 61 significant candidate genes were discovered across 24 tissue types. After multiple testing correction for each tissue, *ADAL* was recurrently identified for the additive and dominant models across multiple tissues. We used a paired WGS and RNA-seq dataset generated from autopsy samples in the current study to illustrate a proof-of-concept. Our approach can be applied to study genetic basis of traits (e.g., transcriptome, proteome, and other phenotype of interests) for animals and plants for which more accurate recombination maps could be generated [42,43]. For instance, molecular mechanisms of breed-defining traits have been characterized in livestock animals by genotyping germline mutations in coding and regulatory sequences [44]. Furthermore, the current approach could be refined to understand how genetic variants in regulatory elements lead to various human phenotypes.

## Figures and Tables

**Figure 1 genes-14-00147-f001:**
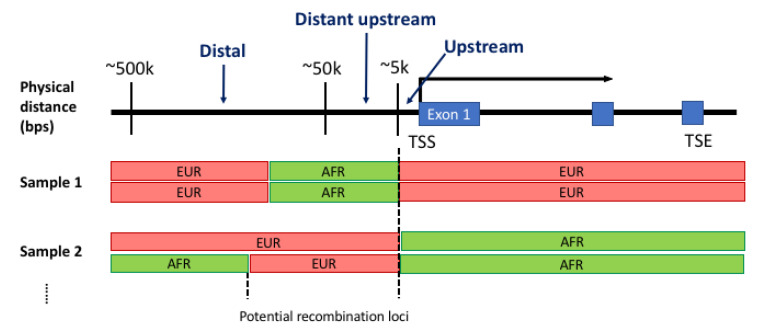
Admixed ancestral structure relative to protein coding gene. Immediate upstream, distant upstream and distal regions are defined by their physical distance from the transcription start site (TSS) for each of protein coding genes defined in the GENCODE annotation. For each individual, a block of genomic region is assigned to one of the continent-level ancestral populations included in the reference panel prepared for RFMix. For the reference panel with two populations—i.e., European (EUR) and African (AFR)—haplotype blocks are assigned to one of the ancestral populations as either heterozygous or homozygous. A transition point between adjacent haplotype blocks suggests ancestral recombination locus.

**Figure 2 genes-14-00147-f002:**
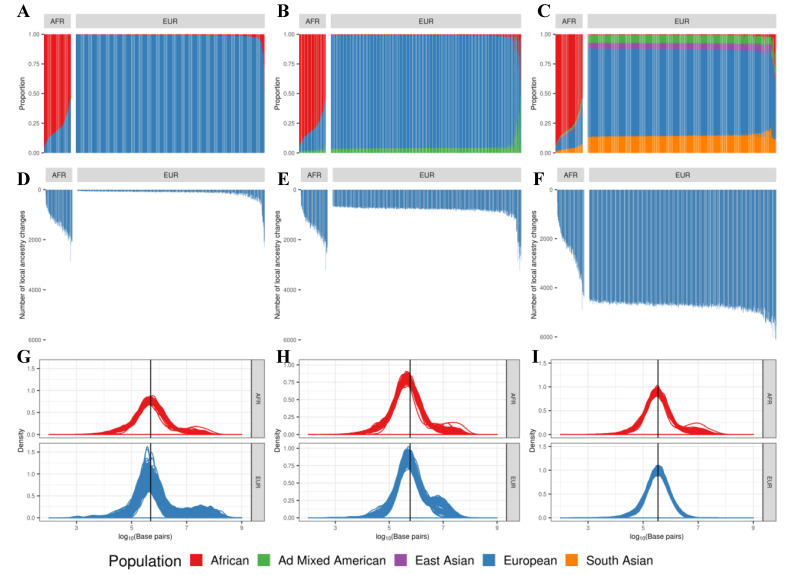
An overview of global and local ancestral structures of whole-genome sequencing data from the Genotype-Tissue Expression project. The overall proportion of continent-level ancestral populations within each subject as predicted by RFMix using the reference panel of (**A**) two populations, (**B**) three populations and (**C**) five populations. The number of transition points within each subject when using two-, three- and five-population panels (**D**–**F**, respectively). The distribution of haplotype block lengths for African and European individuals with two-, three- and five-population panels (**G**–**I**, respectively): African individuals (**top**) and European individuals (**bottom**).

**Figure 3 genes-14-00147-f003:**
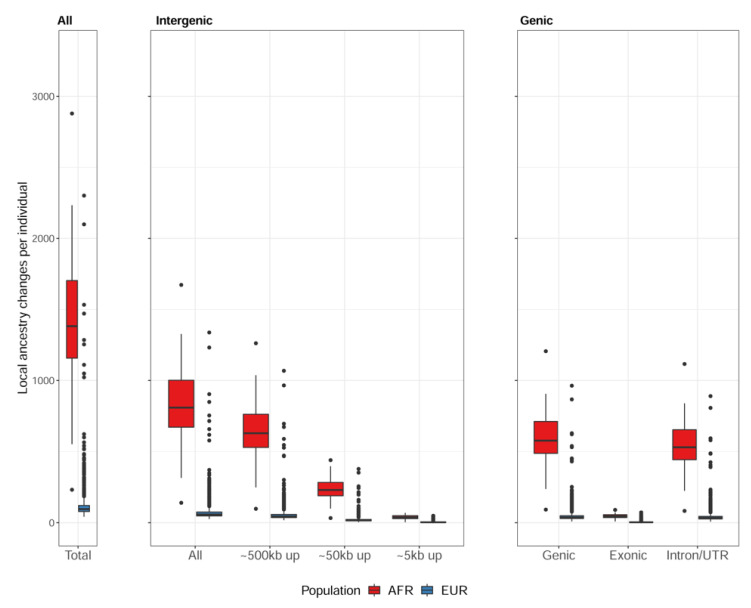
Predicted transition loci relative to the GENCODE gene models: the total number of transition loci from an individual (**left**), the number of transition loci in the intergenic region (**center**) and in the genic region (**right**). African individuals have a significantly larger number of admixed events on average compared to European individuals when the two-population reference panel is used for RFMix analysis for local ancestry inference with transitions.

**Figure 4 genes-14-00147-f004:**
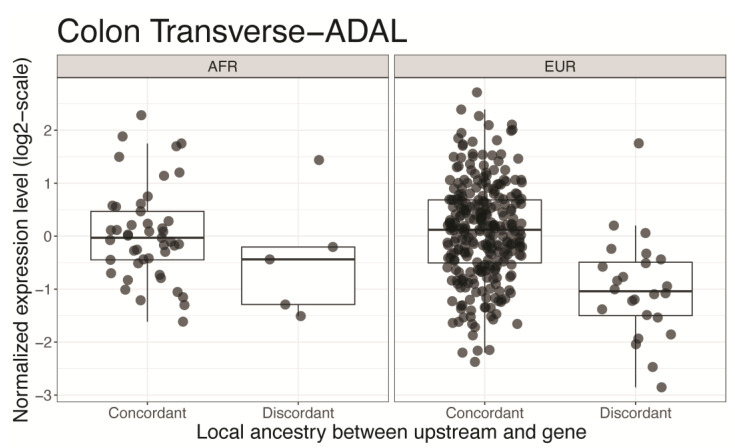
The expression level of the Adenosine Deaminase Like (*ADAL)* gene in transverse colon among the individuals without transition in the upstream (“Concordant”) and among the individuals with transition in the upstream (“Discordant”). The expression level (*y*-axis) shows the value from the gene expression matrix for transverse colon in the GTEx single-tissue *cis*-eQTL data, after normalizing with age, sex, and global ancestry. The left panel shows expression levels for African individuals and the right panel for European individuals. The expression levels are lower among “Discordant” individuals.

**Figure 5 genes-14-00147-f005:**
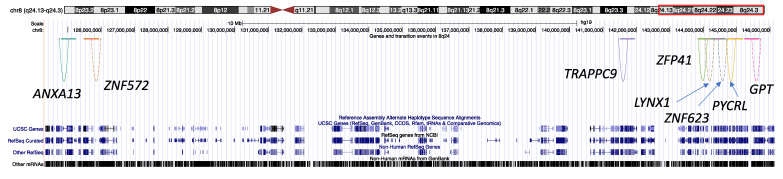
Genes in chromosome 8q24 locus and their upstream regions considered for ancestral structure. Only the genes with significant associations are shown. For each gene, the transcription start site (TSS) that is denoted as a single base position and its upstream region (denoted as a horizontal line in the left/right of TSS) are connected by arcs. For visibility, only the 50~500 kbps upstream regions are shown. The other upstream regions considered during the analysis are located between end points of the arcs. Each gene is represented by different colors and the dashed arcs represent those on the positive strand.

**Table 1 genes-14-00147-t001:** The list of candidate genes across tissue types. The distance ranges are from the transcription start site of gene model to transition points in the upstream. The three models of transition events (i.e., dominant, additive, or recessive) are used for linear regression analysis. False discovery rate is calculated within each model of transition event, distance range, and tissue type.

Model	Distance from Transcription Start Site	Tissue Type	Gene	False Discovery Rate
Dominant	Less than 5 kbps	Small Intestine, Terminal Ileum	*SLC17A9*	0.047
	5~50 kbps	Brain, Cerebellar Hemisphere	*HLA-DMA*	0.018
	50~500 kbps	Adipose, Subcutaneous	*ADAL*	0.00029
			*C10orf107*	0.0049
			*HLA-DQB2*	0.016
		Artery, Aorta	*ADAL*	8.4 × 10^−6^
			*PSORS1C2*	0.007
		Artery, Tibial	*ADAL*	6.6 × 10^−7^
		Brain Cerebellum	*HLA-A*	0.01
		Breast, Mammary Tissue	*ADAL*	0.0035
		Colon, Transverse	*ADAL*	0.00083
		Esophagus, Muscularis	*C10orf107*	0.02
			*ADAL*	0.02
		Heart, Atrial Appendage	*C10orf107*	0.00068
			*PLEK2*	0.04
			*ALOX12*	0.04
		Lung	*PCDHGA6*	0.029
			*PSORS1C2*	0.029
			*STEAP2*	0.03
		Muscle, Skeletal	*HLA-DQB2*	0.025
			*COL8A2*	0.025
		Nerve, Tibial	*ADAL*	2.4 × 10^−6^
		Ovary	*ADAL*	0.0029
		Skin, Not Sun Exposed Suprapubic	*ADAL*	0.035
		Skin, Sun Exposed Lower leg	*ADAL*	0.00068
		Spleen	*ADAL*	0.00055
		Stomach	*ADAL*	0.00074
		Thyroid	*WDR87*	5.5 × 10^−5^
			*ADAL*	0.00012
			*ZSCAN31*	0.0053
		Whole Blood	*MISP3*	0.033
Recessive	Less than 5 kbps	Uterus	*SH3GLB1*	0.043
Additive	Less than 5 kbps	Small Intestine, Terminal Ileum	*SLC17A9*	0.047
	5~50 kbps	Brain, Cerebellar Hemisphere	*HLA-DMA*	0.018
	50~500 kbps	Adipose, Subcutaneous	*HLA-DQB2*	6.6 × 10^−6^
			*ADAL*	0.00014
			*C10orf107*	0.0033
		Adipose, Visceral Omentum	*HLA-DQB2*	0.0092
		Artery, Aorta	*ADAL*	2.1 × 10^−5^
		Artery, Tibial	*ADAL*	6.6 × 10^−7^
		Breast, Mammary Tissue	*ADAL*	0.0074
		Colon, Transverse	*ADAL*	0.00083
		Esophagus, Muscularis	*C10orf107*	0.02
		Heart, Atrial Appendage	*HLA-DQB2*	3.3 × 10^−5^
			*C10orf107*	0.00034
		Heart, Left Ventricle	*HLA-DRB5*	0.015
		Lung	*PCDHGA6*	0.032
			*TLDC1*	0.032
		Muscle, Skeletal	*HLA-DQB2*	3.1 × 10^−7^
		Nerve, Tibial	*ADAL*	3.7 × 10^−6^
		Ovary	*ADAL*	0.0053
		Skin, Not Sun Exposed Suprapubic	*ZNF347*	0.022
			*ADAL*	0.022
		Skin, Sun Exposed Lower leg	*ADAL*	0.00068
		Spleen	*ADAL*	0.00055
		Stomach	*ADAL*	0.00074
		Thyroid	*WDR87*	5.9 × 10^−5^
			*ADAL*	0.00012
			*ZSCAN31*	0.0014
		Vagina	*CTNNA2*	0.029
		Whole Blood	*ZFP57*	0.0029

**Table 2 genes-14-00147-t002:** Significant genes associated with transition events in the regulator region on chromosome 8q24. False discovery rate is calculated within each model of transition event, distance range, and tissue type.

Model	Distance from Transcription Start Site	Tissue Type	Gene	False Discovery Rate
Dominant	5~50 kbps	Skin, Not Sun Exposed Suprapubic	*ANXA13*	0.015
			*TRAPPC9*	0.025
	50~500 kbps	Brain, Spinal Cord Cervical C1	*GPT*	0.035
		Whole Blood	*ZNF572*	0.041
Recessive	Less than 5 kbps	Brain, Anterior Cingulate Cortex	*ZNF623*	0.032
		Brain, Frontal Cortex BA9	*ZNF623*	0.03
		Colon, Transverse	*LYNX1*	0.0015
	5~50 kbps	Brain, Caudate Basal Ganglia	*PYCRL*	0.02
		Brain, Cerebellar Hemisphere	*PYCRL*	0.037
			*TRAPPC9*	0.037
		Brain, Hypothalamus	*PYCRL*	0.024
			*TRAPPC9*	0.024
		Skin, Not Sun Exposed Suprapubic	*TRAPPC9*	0.049
Additive	5~50 kbps	Skin, Not Sun Exposed Suprapubic	*ANXA13*	0.015
			*TRAPPC9*	0.025
	50~500 kbps	Brain, Spinal Cord Cervical C1	*GPT*	0.031
		Skin, Sun Exposed Lower Leg	*ZFP41*	0.025

## Data Availability

Not applicable.

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
