# Peer review of "ADGR: Admixture-Informed Differential Gene Regulation"

_genes, 2023, doi:10.3390/genes14010147_

Round 1
Reviewer 1 Report
This article focused on developing a novel method called "Admixture-informed Differential Gene Regulation (ADGR)" for modeling differences in gene expression. I think such method is a very innovative work and should be very welcome. The work is sound and the method is very attractive in studying the upstream elements and gene expression. I only have a few suggestions.
Minor issue:
1, figure 4, please provide the method and dataset for creating this figure in the legend. Also, please provide the y axis name for the expressions. whether it be log2expression or others.
2, Table2, please provide a figure on the spatial chromosomal structure between the regulator region and the coding sequence. Readers would suspect they are spatially correlated although they are distant in term of physical range.
3, I think this method is very useful and not only applicable in human genomics, please provide a paragraph on its use in animal science and plant science. Please also cite a few articles on animal science and plant science to support this application potential. These could include: Proteome and transcriptome profile analysis reveals regulatory and stress-responsive networks in the russet fruit skin of sand pear | Horticulture Research | Oxford Academic (oup.com) Genotype-specific suppression of multiple defense pathways in apple root during infection by Pythium ultimum | Horticulture Research | Oxford Academic (oup.com)
Based on these comments, I would strongly suggest minor revision.
Reviewer 2 Report
1. They should explain the reason, why average number of transitions points are more in European individuals.
2. Why ADAL expressed different in African and European American Individuals.
3. They should add the allelic level relation.
